# Clinical and cost evaluation of intensive support team (IST) models for adults with intellectual disabilities who display challenging behaviour: a comparative cohort study protocol

Angela Hassiotis,[1] Athanasia Kouroupa  ,[1] Rebecca Jones,[1] Nicola Morant  ,[1] Ken Courtenay,[2] Ian Hall,[3] Vicky Crossey,[4] Renee Romeo,[5] Laurence Taggart,[6] Peter Langdon  ,[7] Victoria Ratti,[1] Vincent Kirchner,[8] Brynmor Lloyd-Evans[1]

For numbered affiliations see end of article.

**Correspondence to**
Athanasia Kouroupa;
athanasia.kouroupa.12@ucl.ac.uk

## ABSTRACT

**Introduction** Approximately 17% of adults with intellectual disabilities (ID) living in the community display behaviours that challenge. Intensive support teams (ISTs) have been recommended to provide high-quality responsive care aimed at avoiding unnecessary admissions and reducing lengthy inpatient stays in England. We have identified two models of ISTs (model 1: enhanced provision and model 2: independent provision). This study aims to investigate the clinical and cost-effectiveness of the two models of ISTs.

**Methods and analysis** A cohort of 226 adults with ID displaying behaviour that challenges who receive support from ISTs from each model will be recruited and assessed at baseline and 9 months later to compare the clinical and cost-effectiveness between models. The primary outcome is reduction in challenging behaviour measured by the Aberrant Behaviour Checklist-Community (ABC-C). The mean difference in change in ABC score between the two IST models will be estimated from a multilevel linear regression model. Secondary outcomes include mental health status, clinical risk, quality of life, health-related quality of life, level of functioning and service use. We will undertake a cost-effectiveness analysis taking both a health and social care and wider societal perspective. Semistructured interviews will be conducted with multiple stakeholders (ie, service users, paid/family carers, IST managers/staff) to investigate the experience of IST care as well as an online survey of referrers to capture their contact with the teams.

**Ethics and dissemination** The study was approved by the London–Bromley Research Ethics Committee (REC reference: 18/LO/0890). Informed consent will be obtained from the person with ID, or a family/nominated consultee for those lacking capacity and from his/her caregivers. The findings of the study will be disseminated to academic audiences, professionals, experts by experience and arm's-length bodies and policymakers via publications, seminars and digital platforms.

**Trial registration number** ClinicalTrials.gov Registry (NCT03586375).

### Strengths and limitations of this study

► This study will be the first systematic national evaluation of intensive support team (IST) models for adults with intellectual disability who display challenging behaviour.
► Accessing the experience of stakeholders (eg, IST managers, staff members, family and paid carers, and service users) will enable an in-depth understanding of how ISTs function and how they respond to people with complex needs.
► This is not a randomised controlled trial so may be subject to biases associated with observational research.

## INTRODUCTION

People with intellectual (also called learning) disabilities (ID), characterised by cognitive delay and difficulties in adaptive functioning, constitute approximately 1% of the population.[1] Seventeen per cent of adults with ID, living in the community, present with serious challenging behaviour including aggression, self-injury or other socially inappropriate behaviours.[2] As many as 100,000 children and adults are estimated to be at risk of admission to inpatient care due to the presence of such behaviours if they are not effectively supported in the community.[3] There are ongoing concerns that these individuals are subject to increased rates of hospitalisation, unnecessary long-term use of psychotropic medication, poorer health, abuse and exclusion.[4]

Intensive support teams (ISTs) are specialist teams which have been advocated for many years as the best services to help people with ID and challenging behaviour remain within their local communities. The teams may

be staffed by one or more professions (eg, psychology, nursing, psychiatry). They deliver interventions such as positive behaviour support, support adults with ID who are in a mental or behavioural crisis, and provide inreach when a person is admitted to a psychiatric inpatient facility.[5 6] ISTs are recommended to provide high-quality proactive and responsive care aimed at avoiding unnecessary admissions or reducing inpatient length of stay and maintaining people in their local community.[6 7] However, there is little evidence to recommend a preferred IST model or way of implementing these aims, and there has not been a comprehensive attempt to describe the clinical and cost-effectiveness of ISTs. Such information would be needed by health funders in order to fund services fit for purpose. The National Institute of Health and Care Excellence Guideline (NG11)[8] reports that: 'It is widely recognised that locally accessible care settings could be beneficial and could reduce costs but there is no strong empirical evidence to support this'.

Previous studies describe either demonstration projects following the closure of institutions[9 10] or region-wide implementation of standalone services.[11] Three small controlled trials have examined, respectively: (1) a standalone specialist-support service delivering standard treatment in addition to behaviour therapy in one area in England[12]; (2) a standalone team delivering assertive outreach for adults with ID who display challenging behaviour in inner London[13] and (3) an active case management model.[14] Comparator treatment was usual care in all three studies. The remit of the ISTs differed between studies, with one providing support to people who display challenging behaviour while the other two provided a replication of the intensive support management from adult mental healthcare. These small studies presented equivocal findings in terms of reduction in challenging behaviour, improved level of functioning and quality of life dependent on the model.

A study[15] showed that positive behavioural outcomes may be achieved by an embedded IST model in which a core team of community ID service staff trained in managing challenging behaviour meet together regularly to discuss referrals with specialist supervision and peer support by the lead clinician. Literature from other population groups (eg, dementia care)[16] and adult mental health[17] suggests that home treatment teams may be effective in managing crises and reducing admissions. Wheeler et al[18] showed that stakeholders in adult mental healthcare have a number of expectations from crisis resolution teams and this is likely to be the case for ISTs in the field of ID. So far, there has been limited reporting on stakeholder experiences of ISTs,[15 18] so little is known whether service users and paid and family carers find the involvement of IST staff and frequency of contact helpful and acceptable.

As part of the Transforming Care programme,[19] the National Health Service (NHS) England has prioritised ISTs as a central pillar in the arsenal of support for individuals who display challenging behaviour, and has invested

in a variety of formats of such teams in areas with high admission rates. The current study is the second phase of a wider project that set out to systematically examine the implementation of ISTs in England.

The first phase identified two IST service models[20]: enhanced provision (model 1) and independent provision (model 2) following an England-wide survey of ISTs (n=73).[20] The components of each model as reported by professionals in the survey are shown below (table 1).

There is limited evidence for the comparative clinical and cost-effectiveness of either enhanced or independent provision[15] and of long-term clinical gains. Furthermore, satisfaction of stakeholders with the ISTs has hardly featured in the literature and that deserves better understanding if the ISTs are fit for purpose and responsive to patient need.

**Table 1** Characteristics of IST models

| ISTs | |
| --- | --- |
| **Model 1—enhanced provision** | **Model 2—independent provision** |
| Integrated within the broader Community Intellectual Disability Services | Separate from Community Intellectual Disability Services |
| Long-term support (more than 6 months) | Short-term support (3–12 months) |
| Accept self-referrals | Referral via professionals |
| Large caseload (20+) | Small caseload (up to 15) |
| Less likely to use outcome measures | Use of outcome measures |

IST, intensive support team.

## OBJECTIVES

The overall aim of phase 2 of the study is to investigate patient outcomes (eg, challenging behaviour, risk, adaptive behaviour, service use, quality of life) and service-level outcomes (eg, engagement with the IST, satisfaction with referral process, number of referrals, experience of care provided).

### Specific objectives

► To generate evidence regarding which IST model best supports improved outcomes for adults who display challenging behaviour.
► To estimate the costs of the two IST models and examine cost-effectiveness.
► To explore views and experiences of ISTs among relevant stakeholders (service users, paid and family carers, and service providers).
► To investigate how ISTs impact the lives of adults with IDs who display challenging behaviour in their caseload, their families and the local services.

Findings will generate evidence to inform and support decision-making on commissioning of ISTs for adults with ID who display challenging behaviour.

## METHODS AND ANALYSIS

The research team will compare the two models, enhanced provision and independent provision, respectively, of ISTs already identified in phase 1 of the study.[20] The research assistant or clinical study officers will collect study participant-level outcomes at two assessment points (baseline and 9 months) and service-level outcomes over the same time period. The research assistant will also collect qualitative data to understand the experiences and views of key stakeholders, and stakeholder experiences of the different models.

### Setting and participants

Eight enhanced service provision ISTs and eight independent service provision ISTs (table 1) will be selected at random from ISTs surveyed in phase 1 of the study.[20] Service users, either already under the care of the 16 ISTs or newly referred, and their caregivers will be identified by IST staff either at first clinical assessment or from the IST services caseloads and asked if they are interested in taking part in the study. Following an expression of interest, the researchers will contact them to provide information about the study and to obtain written or audio-recorded informed consent from those agreeing to undergo further research assessments.

### Inclusion criteria

1. Service: IST adheres to one of the identified models (table 1); has been operational for at least 12 months; there is commitment to fund it for the study duration and it can achieve the sample size estimates.
2. Service users: receive support from IST service; mild to profound ID based on clinical diagnosis; aged 18 years and over.

### Exclusion criteria

1. Service: has been operational for less than 12 months at selection or there are plans to dissolve it.
2. Service users: primary clinical diagnosis of personality disorder or substance misuse; decision by clinical team a referral to the study would be inappropriate (eg, due to ongoing legal challenges to the service or the person being acutely unwell).

### Consent

Consent will be sought from service users and their respective carers (ie, paid and family) and/or nominated consultees where the participant with ID lacks capacity in accordance with the Mental Capacity Act 2005.[8] For those service users with decision-making capacity, the researcher will speak to the potential participant by phone or in person and will give or send the service user participant information sheet (in Easy Read format) to inform them of the study. If the service user agrees to take part in the study, the individual will complete a written consent form or consent will be taken by telephone and audio-recorded for purposes of verifying consent. The researcher will then repeat the above process (using the carer participant information sheet) with the service users' paid or family carer to seek their consent to take part and complete the rest of the study measures. For those service users lacking decision-making capacity, the researcher will approach the personal or nominated consultee for that person (using the consultee information sheet) and seek written or audio-recorded advice about including the service user in the study.

### Outcome measures

#### Study participant-level outcomes

Each participant will be recruited at any point of IST care and will be assessed at baseline and at 9 months which reflects the time period an IST may be expected to be involved in working with a participant. This length of follow-up was chosen because our research and clinical experience indicates that, while there is a range of duration for IST involvement, assessment and delivery of behaviour support plans usually require about 3 months to complete and adults with ID and challenging behaviour often remain in the care of the IST for approximately 12 months.

The primary outcome is challenging behaviour measured by the Aberrant Behaviour Checklist-Community (ABC-C)[21] which has been widely used in the target population and is completed by the caregiver (eg, family, paid care home staff, IST staff).

A range of secondary endpoints examine the following: the carer-reported Psychopathology Assessment for Adults with Developmental Disabilities checklist [22] is a mental health screening tool which provides sufficient information on potential mental health comorbidity that is often underascertained in adults with ID. The Threshold Assessment Grid[23] measures clinical risk in adults with ID. The Quality of Life (QoL)[24] will be used with the individual with ID and/or the caregiver, if participants lack capacity, to measure the individual's quality of life. The Adaptive Behaviour Scale-Short Form assesses level of functioning.[25] The Client Service Receipt Inventory (CSRI)[26] (adapted for the study, 6-month retrospective service use at each assessment point) is a widely used service-use questionnaire and has been validated for use in mental health and ID services research.

Health-related quality of life will be measured by the EQ-5D-5L.[27] The EQ-5D is used to generate quality-adjusted life years (QALYs). The self-report version will be used with the individual with ID where possible, and the proxy version will be used with the caregiver in the cases of participants lacking capacity.

The researchers will collect sociodemographic information, as well as number of hospital admissions and maintenance of accommodation at follow-up.

The study participant flow chart is shown in figure 1.

### Service-level outcomes

The researchers will collect data from all participating ISTs including the number of people referred; proportion who engage with IST; time to first assessment and delivery of management plan; other IST scope (eg, days

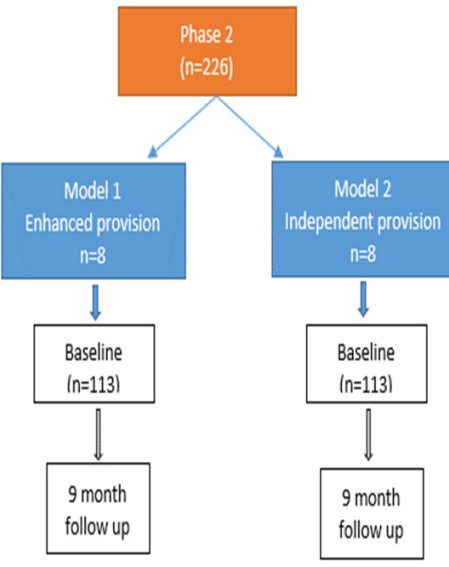

**Figure 1** Study participant flow chart.

of training given) and other engagement with local services (eg, joint assessments with crisis teams) that will provide additional information about the ISTs' caseload size. We will map our service data onto the reports from the Mental Health and Learning Disabilities Data Set over the study period which provides information on hospital admissions and compares with admissions by IST model. This will provide a proxy measure of IST model impact on admissions.

### Referrer's questionnaire

Practitioners from community ID services who frequently refer to ISTs across all sites will be invited to complete a brief questionnaire designed to capture satisfaction with referral processes, reasons for referrals and number of referrals to IST. The questionnaire will be administered either online, via email or post, or a combination of these.

### Qualitative substudy: stakeholder's views of ISTs

An embedded qualitative substudy will allow to investigate how IST care is experienced by service users and family carers, and to obtain a multiperspective view of IST functioning within local service contexts, based on user, carer and practitioner views. The researchers will conduct semistructured interviews with (1) IST managers and staff, (2) service users, and (3) family and paid carers. Up to eight service users per IST model, including those who may need support to communicate (provided by family or paid carers); and up to eight carers (a mixture of family and paid) per IST model will be included. The researchers will also attempt to interview a small number of family carers who have recently declined IST contact. Purposive sampling will be employed, in order to include a range of ages, work experience, professional backgrounds, and carer roles and relationships in our sample.

In order to gain a broad perspective on IST implementation in England, the researchers will aim to conduct interviews with all managers of the IST services taking part in the phase 2. Interviews with IST managers and staff will be conducted in the early stages of the research study; interviews with service users and family and paid carers will be conducted near the time of discharge from IST services or around the 9-month follow-up, whichever is sooner.

The interview will take place at a location convenient (ie, home, clinic or remotely with the aid of technology) for the participant and will be audio-recorded. Across all stakeholder groups, interview schedules will be designed to explore views and experiences of the role and functioning of ISTs, and how they interface with other health and social care provision within their local context. The semistructured interview schedule will address the operational functioning, benefits and limitations of ISTs, and explore the factors that might affect these following consultation with the patient and public involvement (PPI) representatives. In order to link stakeholder perspectives to specific service contexts, we aim to carry out data collection within seven IST 'case study' sites. These sites will be selected from the larger pool of the 16 services, with an aim to include services that vary in terms of team size, caseload size and styles of working based on the reflections and other relevant information available to the research team. Four services will be selected as case studies of 'independent' ISTs and three services as case studies of 'enhanced model' ISTs. The use of case study sites is appropriate where the research focus is to assess 'mega systems' of great complexity.[28]

### Study termination

The end of the study will be the date of the 9-month follow-up assessment with the study participant. For those who take part in interviews, these will be arranged to take place shortly after the questionnaire completion. Further, if the study failed to recruit adequately, it would be discussed with the study Steering Committee and the funders as to whether it should stop or be extended.

### Sample size

Based on two IST models, it was estimated that a sample of 96 participants per model (192 in total) would be required to detect a difference of 0.45 SDs in primary outcome score between the two models at the 5% significance level with 80% power and assuming an intraclass correlation of 0.02.[12 29] After factoring in 15% loss to follow-up, the estimated sample size required increased to 113 participants per model (226 participants in total).

### Statistical analysis plan

As per the predetermined statistical analysis plan, the baseline characteristics of participants will be summarised separately for the two IST models. Categorical variables will be reported as counts and percentages, while continuous variables will be summarised as means and SDs or

medians and IQRs as appropriate depending on the distribution of the data.

The primary outcome is challenging behaviour measured by total score on the ABC. Change in ABC total score from baseline to 9-month follow-up will be calculated for each participant. The mean difference in ABC score between the two IST models will be estimated from a multilevel linear regression model with change in ABC score as the outcome, a fixed effect of model as the main exposure, and a random effect of IST to account for clustering within services.

Age, gender, living arrangements, level of ID, level of risk, presence of autism and clinical comorbidities have been identified as potential confounders and will be specified as covariates in adjusted multilevel linear regression models similar to the main unadjusted model described above. The estimated difference in change in ABC total score from both unadjusted and adjusted models will be reported with accompanying 95% CI and p value.

The secondary outcomes detailed above will be analysed using statistical models analogous to those for the primary outcome. Binary outcomes will be analysed using multilevel logistic regression models. Non-normally distributed continuous outcomes will be analysed using the bootstrap method and bias corrected 95% CI will be reported. P values will not be reported for secondary outcomes.

### Economic evaluation plan

As per the predetermined health economic analysis plan, we shall derive and report the costs of each IST service model over 9 months. To calculate the cost of each IST service model, we will use information on the description, time inputs of staff and related caseloads of professionals in the IST.

The cost-effectiveness analyses will be conducted from a health and social care perspective and a wider societal perspective. Data on health and social care services and support provided by statutory services external to the ISTs will be obtained from the CSRI covering a retrospective period of 6 months. To service use and support data, we shall attach unit costs taken from a wide range of sources.[30 31] Costs of unpaid care will be estimated from information on volume and type of support, the opportunity cost of lost work (wage rate) for carers in paid employment and replacement cost for those not in paid employment based on cost of a home care worker. We shall extrapolate the 6-month costs over the 9-month period and examine the link between costs and sociodemographic, clinical characteristics of individuals in the study.

The cost-effectiveness of one IST model over another will be compared by calculating incremental cost-effectiveness ratios, defined as difference in mean costs divided by difference in mean effects. Cost-effectiveness acceptability curves[32] will be plotted for each cost–outcome combination to show the likelihood of one IST model as cost-effective relative to another for a range of (implicit) values placed on incremental outcome improvements. Using the net benefit (NB) approach, monetary values of incremental effects and incremental costs are combined, and NB derived as: $NB = \lambda \times (\text{effect}_b - \text{effect}_a) - (\text{cost}_b - \text{cost}_a)$. Where, $\lambda$ is the willingness to pay for a unit improvement in effectiveness (ABC, QALYs and QoL) and subscripts 'a' and 'b' denote *IST model a* and *IST model b*, respectively. This approach allows costs and outcomes to be considered on the same monetary scale, taking account of sampling uncertainty and adjusting for baseline covariates and clustering.

### Qualitative analysis plan

Interviews will be audio-recorded and transcribed verbatim by an external agency. Data will be analysed using thematic analysis[33] conducted using NVivo software. A staged, collaborative and primarily inductive analytical approach will be adopted, allowing us to iteratively develop a set of themes to capture key concerns and topics, as well as more abstract or underlying issues. Although numbers in stakeholder groups linked to each IST model may be relatively small, triangulation of the various stakeholder perspectives will allow us to obtain a broad picture of each IST model. Thus, we will be able to compare the IST models in terms of multiple stakeholders' views, as well as analysing the data set as a whole to understand broadly common views and experiences of ISTs. Analysis will involve close collaboration between study researchers, the qualitative lead and other key members of the study team. Service user and carer group meetings will be scheduled three times per year. The members of the advisory groups will receive easy read summary documents to be informed about the study progress, share views on different recruitment strategies, comment on the coding, analysis and interpretation of study findings.

### Patient and public involvement

A service user reference group (Camden Disability Action) has supported the project and assisted with the research governance requirements including input to easy read versions of study materials (ie, information sheets, consent forms) and piloting of instruments and qualitative interview topic guides with reference to people with ID. Family carer representatives from the Challenging Behaviour Foundation, an independent national charity of family carers of people with IDs and challenging behaviour, have also been involved in the conduct of the study and advising on recruitment as well as reviewing study-related information including the qualitative interviews topic guides. Meetings with both expert by experience groups are scheduled to take place every 3 months. Finally, there is service user and family caregiver representation at the study Steering Committee overseeing the study.

### ETHICS

Ethical approval for this study was granted on 26 June 2018 by the London Bromley Research Ethics

Committee (REC) (18/LO/0890). All personal data are handled in accordance with the Data Protection Act. All outcome measures are stored electronically, and access is restricted to the principal investigator of the study and nominated researchers. All identifiable documents are saved as password-protected files on password-protected computers. Confidential information (including signed consent forms) are stored separately from paper records (eg, Clinical Research Forms) in locked filing cabinets in a restricted area accessed by members of the University College London, Division of Psychiatry. Data management and data quality checks will be performed by the research study team.

## Dissemination plan

Our dissemination plan was developed in close collaboration with PPI representatives. Our dissemination and impact activities will include open-access academic publications in high-impact journals, presentations at academic and non-academic conferences, hosting a free public webinar about the study and producing briefings for carer groups, charities and NHS services. A newsletter about the study progress will be sent to participants every 6 months and also available on the study website (https://www.ucl.ac.uk/psychiatry/research/epidemiology/pis/hassiotis-research-portfolio/intensive-support-teams). The research team will submit manuscripts to peer-reviewed publications, and authorship inclusion and order will be guided by levels of contribution. All publication material will acknowledge the funding contribution from the National Institute for Health Research. Requests for access to the anonymised data and statistical code should be addressed to the corresponding author once the study funder report and main papers are published.

## Study progress and COVID-19 impact

The first phase of the study has been completed and reported.[20] The study has faced recruitment challenges which in the main included: reduced capacity to identify sufficient numbers of eligible participants due to small and static caseloads, services withdrawing or changing their remit halfway through the study, inadequate communication between ISTs and Clinical Research Networks which hampered research support. The number of ISTs was expanded to 23, 21 of which contributed participants to the study. The originally estimated 9 months of recruitment were extended to 17 with the last participant enrolled in May 2020; 180 9-month follow-ups have been completed. The last 4 months of recruitment occurred during the coronavirus pandemic and the subsequent lockdown which necessitated changes in study procedures including the introduction of remote completion of the research assessments and qualitative interviews. Those changes were non-substantial amendments (number 7) to the REC that had originally approved the study. Thirteen participants have been lost to follow-up with one death from COVID-19.

**Author affiliations**
[1]Division of Psychiatry, University College London, London, UK
[2]Assessment and Intervention Team, Barnet Enfield and Haringey Mental Health NHS Trust, London, UK
[3]Hackney Integrated Learning Disability Service, East London NHS Foundation Trust, London, UK
[4]South West Community Learning Disability Team & Mental Health Intensive Support and Treatment Team, NHS Lothian, Edinburgh, UK
[5]Institute of Psychiatry, Psychology and Neuroscience, King's College London, London, UK
[6]Institute of Nursing and Health Research, University of Ulster, Antrim, UK
[7]Centre for Educational Development, Appraisal and Research, University of Warwick, Coventry, UK
[8]Medical Director, Camden and Islington NHS Foundation Trust, London, UK

**Contributors** AH conceived the study, wrote the funding application and led the study design along with NM, RJ, KC, VC, IH, RR, LT, PL, VK and BL-E who were the funding co-applicants. RJ developed the statistical element of the study. RR developed the health economic element of the study. NM developed the qualitative analysis section of the study. VR drafted the first version of the protocol manuscript for publication with input from AH and RR. NK finalised the protocol paper and liaised with all authors. All authors approved the final protocol.

**Funding** The study is funded by the National Institute of Health Research (NIHR) Health Services and Delivery Research (HS&DH) programme (project reference: 16/01/24). This publication presents independent research funded by the NIHR.

**Disclaimer** The views expressed are those of the author(s) and not necessarily those of the NHS, the NIHR or the Department of Health and Social Care.

**Competing interests** None declared.

**Patient and public involvement** Patients and/or the public were involved in the design, or conduct, or reporting, or dissemination plans of this research. Refer to the Methods section for further details.

**Patient consent for publication** Not required.

**Provenance and peer review** Not commissioned; externally peer reviewed.

**ORCID iDs**
Athanasia Kouroupa http://orcid.org/0000-0003-3659-160X
Nicola Morant http://orcid.org/0000-0003-4022-8133
Peter Langdon http://orcid.org/0000-0002-7745-1825

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
