## [Reviewer comments · BMJ Open]

ARTICLE DETAILS

TITLE (PROVISIONAL)	Clinical and cost evaluation of intensive support team (IST) models for adults with intellectual disabilities who display challenging behaviour: a comparative cohort study protocol
AUTHORS	HASSIOTIS, ANGELA; Kouroupa, Nancy; Jones, Rebecca; Morant, Nicola; Courtenay, Ken; Hall, Ian; Crossey, Vicky; Romeo, Renee; Taggart, Laurence; Langdon, Peter; Ratti, Victoria; Kirchner, Vincent; Lloyd-Evans, Brynmor

VERSION 1 – REVIEW

REVIEWER	AZIDAH ABDUL KADIR Universiti Sains Malaysia Malaysia
REVIEW RETURNED	21-Sep-2020

GENERAL COMMENTS	A very good and comprehensive study that analyse cost effectiveness of the intervention programs for the adults with i disabilities
---

REVIEWER	Emily Roberts-Tyler Collaborative Institute of Education Research, Evidence and Impact, School of Education and Human Development, Bangor University, Wales, UK
REVIEW RETURNED	27-Sep-2020

GENERAL COMMENTS	Thank you for the opportunity to review the protocol for this important study comparing the clinical and cost effectiveness of two different models of Intensive Support Teams for adults with ID displaying behaviour that challenges. The study is well designed to address the research questions, and a thorough protocol is reported of both quantitative and qualitative aspects of the evaluation. Further, there is a clear consideration of the increased likelihood of the need for remote consent in the current climate. Some clarifications and minor edits to consider: An interesting issue was raised in the introduction regarding the lack of evidence for long-term outcomes of ISTs. Do any aspects of this study address the lack of evidence for long-term outcomes? Although measures are conducted at baseline and 9 months, this is still during engagement with the IST and does not seem to shed light on whether potential gains would maintain after discharge or transfer. It would be useful to clarify whether or not this is the case, or whether future studies would be investigating this aspect.
---

	Page 4 Line 26-28 – rephrase sentence for clarity (is it just PBS that is considered the intervention, whereas the others are general services provided?) Line 52 – Clarify what is meant by ‘delivering Applied Behaviour Analysis’; Applied Behaviour Analysis itself is not an intervention. Page 5/6 Line 60 + Lines 1-2 – clarify whether these issues of disruption of dissatisfaction relate to a specific model of ISTs, or whether they are general concerns relevant to both. Page 8 Line 24 – change ‘assess’ to ‘assesses’. Line 55 – consider ‘We will map our service data’ for consistency Line 58 – changed ‘compared’ to ‘compares’ Page 9 Lines 28-29 – It would be useful to explain the rationale for the case studies including four services for the “independent” IST model and three services for the “enhanced” model. Page 10 Sample size section – It would be useful to provided clarification of the basis for the sample size estimates. Page 12 Lines 21-22 – It would be interesting to have an indication of what this consultation will look like and how it will feed into the reported results of the study. Page 13 Line 44 – typo (months) Line 49 – typo (either ‘Thirteen participants’ or ‘Thirteen of the participants’)
--	--

REVIEWER	Dr Judy Clegg Human Communication Sciences Health Sciences School University of Sheffield
REVIEW RETURNED	14-Oct-2020

GENERAL COMMENTS	Many thanks for asking me to review this protocol which I read with much interest. It is a really interesting and innovative study. It is also very welcome given the lack of research into effective interventions in this population. I do commend the team taking this on as it must be incredibly difficult, particularly in the COVID-19 context. In line with a review of a protocol, I do seek some clarification, as follows:  1. The two IST models being evaluated: there needs to be a clearer distinction made between the two models. There is some good description on page 7 including the table (which is very helpful). However, there needs to more description as to the differences between the two. It isn't clear as to why 'enhanced' is 'enhanced' and why this is 'better' than independent. It may seem obvious for those very familiar with these models but many readers won't be. 2. There needs to be more explicit rationale re. the IST models as 'intervention'. This is a tricky question in this field of pragmatic/behaviour type interventions versus medical interventions which are practically easier to evaluate through RCTs and so on. It just needs more explicit description as to why
---

	these models are interventions and therefore seek to be effective in changing outcomes for the person with ID. 3. Can it be explained why there is not a 'treatment' as usual type model - or is this subsumed by the independent IST model? If not, then I think this needs to be more explicit. 4. The primary and secondary outcome measures are explained. Could there be some further rationale as to why hospital admissions (i.e., a reduction of) is not the primary outcome measure and challenging behaviour is? It seems to be that the primary aim of the IST models is to reduce hospital admissions and not to reduce challenging behaviours per se. I'm sure I've misunderstood this so please could the rationale be clearer on this. 5. On page 8, could it be made clearer who is completing the report measures which are the secondary outcome measures. I assume it is the same person/people as for the primary outcome measure. 6. I like the addition of the qualitative study as this will add rich data to the study that the other measures won't capture. In terms of study protocols, qualitative methods don't usually feature in a study protocol so I think the rationale for a mixed methods approach needs some further rationale. Could the selection of the case study sites here be explained further. It wasn't clear to me why and how these were chosen. Could there be more description of the interview schedules as well in terms of number/type of questions and if they were piloted and if so, the findings from the piloting and any changes made to the schedule as a result. Finally, I think it would be helpful to readers to update the section at the end as to how the study is progressing given the current COVID-19 context. My experience of these services at the moment is that they are restricted in terms of accessing homes/patients and so on so I'm sure this must be impacting on the study at this time. I do hope the above is constructive.
--	--

VERSION 1 – AUTHOR RESPONSE

Reviewer: 1

Comments to the Author

A very good and comprehensive study that analyse cost effectiveness of the intervention programs for the adults with i disabilities

Our answer: Thank you for your comment.

Reviewer: 2

Comments to the Author

Thank you for the opportunity to review the protocol for this important study comparing the clinical and cost effectiveness of two different models of Intensive Support Teams for adults with ID displaying behaviour that challenges. The study is well designed to address the research questions, and a

thorough protocol is reported of both quantitative and qualitative aspects of the evaluation. Further, there is a clear consideration of the increased likelihood of the need for remote consent in the current climate. Some clarifications and minor edits to consider: An interesting issue was raised in the introduction regarding the lack of evidence for long-term outcomes of ISTs. Do any aspects of this study address the lack of evidence for long-term outcomes? Although measures are conducted at baseline and 9 months, this is still during engagement with the IST and does not seem to shed light on whether potential gains would maintain after discharge or transfer. It would be useful to clarify whether or not this is the case, or whether future studies would be investigating this aspect.

Our answer: Thank you for your comments. Below are our responses (tracked changes manuscript):

Comment about remote consent

There's clarification already that written, or audio recorded informed consent will be obtained from participants on page 7 line 10, 31, page 14 line 2).

Comment about long term outcomes

Clarification has been added on page 6 line2 and 8-9.

Page 4

Line 16-18 – rephrase sentence for clarity (is it just PBS that is considered the intervention, whereas the others are general services provided?)

Our answer: We have not changed the sentence as PBS is widely considered to be a multimodal complex intervention.

Line 52 – Clarify what is meant by 'delivering Applied Behaviour Analysis'; Applied Behaviour Analysis itself is not an intervention.

Our answer: Clarification point has been added on page 4 line 31.

Page 5/6

Line 60 + Lines 1-2 – clarify whether these issues of disruption of dissatisfaction relate to a specific model of ISTs, or whether they are general concerns relevant to both.

Our answer: Clarification point has been added on page 6 line 2 and 8-9.

Page 8

Line 24 – change 'assess' to 'assesses'.

Line 55 – consider 'We will map our service data' for consistency

Line 58 – changed 'compared' to 'compares'

Our answer: Thank you for the grammar comments. All amended (see page 8 line 23, page 9 line 9, 11).

Page 9

Lines 28-29 – It would be useful to explain the rationale for the case studies including four services for the "independent" IST model and three services for the "enhanced" model.

Our answer: Clarification and literature has been added about the use of case studies on page 11 line 1-7).

Page 10

Sample size section – It would be useful to provided clarification of the basis for the sample size estimates.

Our answer: The sample size estimation has been explained. Additional reference has been added on page 10 line 18 that further elaborates the sample size estimations.

Page 12

Lines 21-22 – It would be interesting to have an indication of what this consultation will look like and how it will feed into the reported results of the study.

Our answer: A clarification point was added on pg 13 line 14-17.

Page 13

Line 44 – typo (months)

Line 49 – typo (either 'Thirteen participants' or 'Thirteen of the participants')

Our answer: Thank you for your grammar comments. All amended on page 14 line 27, 32.

Reviewer: 3

Comments to the Author

Many thanks for asking me to review this protocol which I read with much interest. It is a really interesting and innovative study. It is also very welcome given the lack of research into effective interventions in this population. I do commend the team taking this on as it must be incredibly difficult, particularly in the COVID-19 context. In line with a review of a protocol, I do seek some clarification, as follows: 1. The two IST models being evaluated: there needs to be a clearer distinction made between the two models. There is some good description on page 7 including the table (which is very helpful). However, there needs to more description as to the differences between the two. It isn't clear as to why 'enhanced' is 'enhanced' and why this is 'better' than independent. It may seem obvious for those very familiar with these models but many readers won't be. 2. There needs to be more explicit rationale re. the IST models as 'intervention'. This is a tricky question in this field of pragmatic/behaviour type interventions versus medical interventions which are practically easier to evaluate through RCTs and so on. It just needs more explicit description as to why these models are interventions and therefore seek to be effective in changing outcomes for the person with ID. 3. Can it be explained why there is not a 'treatment' as usual type model - or is this subsumed by the independent IST model? If not, then I think this needs to be more explicit. 4. The primary and secondary outcome measures are explained. Could there be some further rationale as to why hospital admissions (i.e., a reduction of) is not the primary outcome measure and challenging behaviour is? It seems to be that the primary aim of the IST models is to reduce hospital admissions and not to reduce challenging behaviours per se. I'm sure I've misunderstood this so please could the rationale be clearer on this. 5. On page 8, could it be made clearer who is completing the report measures which are the secondary outcome measures. I assume it is the same person/people as for the primary outcome measure. 6. I like the addition of the qualitative study as this will add rich data to the study that the other measures won't capture. In terms of study protocols, qualitative methods don't usually feature in a study protocol so I think the rationale for a mixed methods approach needs some further rationale. Could the selection of the case study sites here be explained further. It wasn't clear to me why and how these were chosen. Could there be more description of the interview schedules as well in terms of number/type of questions and if they were piloted and if so, the findings from the piloting and any changes made to the schedule as a result.

Finally, I think it would be helpful to readers to update the section at the end as to how the study is progressing given the current COVID-19 context. My experience of these services at the moment is that they are restricted in terms of accessing homes/patients and so on so I'm sure this must be impacting on the study at this time. I do hope the above is constructive.

Our answers:

1. A reference has been added to direct the reader get detailed information about the two models on page 5 line 20-22).
2. Information about ISTs has been provided in the manuscript (see page 4 line 13-16). However, it is not our intention here to provide a full historical context of the ISTs although such information has been added as references (see ref 6-7 on page 4 line 20).
3. This is not an interventional study. It's an observational study where we monitor the (clinical and cost) effectiveness of support provided to people with intellectual disabilities and carers by ISTs across England. The support each IST provides is treatment as usual but the operation of each IST (e.g., size and type of referrals, caseload, the number of professionals etc.) differ, as shown in table 1 page 5.
4. At the time of the study and even now, we did not have enough information about what a reasonable reduction in admissions might be. We therefore, used reduction in challenging behaviour as a proxy measure of the impact that ISTs might have and which could lead to reductions in admissions. We are collecting information about admissions and it will be a variable in sensitivity analyses.
5. The relevant sections have been updated throughout the manuscript. The research team (including research assistant or clinical study officer) is administering all measures with carers paid or family; however, it is not always possible to have the same respondent given that both parents as well as paid carers may be unavailable or have left their posts. We, however, try to seek a respondent that knows the individual with intellectual disability for at least 6 months.
6. We are not familiar with the reviewer's standpoint vis a vis the qualitative elements of a mixed methods study. We have added further information in the text which we hope covers this point. In relation to the interview schedules these are all standardised psychometrically valid and whilst the researchers were trained in their completion. No changes can be made to the questionnaires and data quality checks indicate that there was no systematic error in completion of the items included. A few more information have been added on pg 10 lines 17-19, 21-24.
7. COVID-19 section: We have made reference to challenges that the study faced during the pandemic (see page 14); we did not carry out any interventions, whether participants received or not support will be recorded in the Client Service Receipt Inventory and quality of life questionnaire.

VERSION 2 – REVIEW

REVIEWER	Emily Roberts-Tyler School of Education and Human Sciences
REVIEW RETURNED	18-Jan-2021
GENERAL COMMENTS	The authors have very clearly addressed all the comments in my earlier review. I look forward to following the outcomes of this research.